# Genetic Diversity in Invasive Populations of Argentine Stem Weevil Associated with Adaptation to Biocontrol

**DOI:** 10.3390/insects11070441

**Published:** 2020-07-14

**Authors:** Thomas W. R. Harrop, Marissa F. Le Lec, Ruy Jauregui, Shannon E. Taylor, Sarah N. Inwood, Tracey van Stijn, Hannah Henry, John Skelly, Siva Ganesh, Rachael L. Ashby, Jeanne M. E. Jacobs, Stephen L. Goldson, Peter K. Dearden

**Affiliations:** 1Genomics Aotearoa and Department of Biochemistry, University of Otago, Dunedin 9054, Aotearoa, New Zealand; lelma868@student.otago.ac.nz (M.F.L.L.); shannon.elisa.taylor@gmail.com (S.E.T.); sninwood@gmail.com (S.N.I.); skejo931@student.otago.ac.nz (J.S.); peter.dearden@otago.ac.nz (P.K.D.); 2AgResearch, Grasslands Research Centre, Palmerston North 4410, New Zealand; Ruy.Jauregui@agresearch.co.nz (R.J.); sivaganesh1955@outlook.com (S.G.); 3AgResearch, Invermay Agricultural Centre, Private Bag 50034, Mosgiel, New Zealand; tracey.vanstijn@agresearch.co.nz (T.v.S.); hannah.henry@agresearch.co.nz (H.H.); rachael.ashby@agresearch.co.nz (R.L.A.); 4AgResearch, Lincoln Science Centre, Private Bag 4749, Christchurch 8140, New Zealand; Jeanne.jacobs@agresearch.co.nz (J.M.E.J.); stephen.goldson@agresearch.co.nz (S.L.G.); 5Bio-Protection Research Centre, PO Box 85084, Lincoln University, Lincoln 7647, New Zealand

**Keywords:** biological control, invasive species, argentine stem weevil, population genetics, genotyping-by-sequencing

## Abstract

Modified, agricultural landscapes are susceptible to damage by insect pests. Biological control of pests is typically successful once a control agent has established, but this depends on the agent’s capacity to co-evolve with the host. Theoretical studies have shown that different levels of genetic variation between the host and the control agent will lead to rapid evolution of resistance in the host. Although this has been reported in one instance, the underlying genetics have not been studied. To address this, we measured the genetic variation in New Zealand populations of the pasture pest, Argentine stem weevil (*Listronotus bonariensis*), which is controlled with declining effectiveness by a parasitoid wasp, *Microctonus hyperodae*. We constructed a draft reference genome of the weevil, collected samples from a geographical survey of 10 sites around New Zealand, and genotyped them using a modified genotyping-by-sequencing approach. New Zealand populations of Argentine stem weevil have high levels of heterozygosity and low population structure, consistent with a large effective population size and frequent gene flow. This implies that Argentine stem weevils were able to evolve more rapidly than their biocontrol agent, which reproduces asexually. These findings show that monitoring genetic diversity in biocontrol agents and their targets is critical for long-term success of biological control.

## 1. Introduction

Biological control of pests via the release of specialist natural predators can provide continued, self-sustaining, non-polluting and inexpensive management. Although the estimated chance of establishment of biocontrol programs is low [1], biocontrol agents maintain their efficacy once established [2], partly because the agent can evolve adaptations to counter adaptations in the host [3,4].

A biocontrol system has been in use since the 1990s to manage a destructive, invasive pest of New Zealand pastures, the Argentine stem weevil (ASW; *Listronotus bonariensis* Kuschel) (Coleoptera: Curculionidae). New Zealand pastures are highly modified, based on a very low number of introduced Palearctic plant species, and are particularity susceptible to pest impacts [5]. This susceptibility is due to low plant and animal diversity, resulting in low biotic resistance to invasive species [5]. In New Zealand, adult ASW populations can reach densities of 700 adults m-2 and cause economic impacts of up to NZ$200M per annum [6,7,8]. Conventional chemical control of ASW is ineffective, environmentally damaging and uneconomical (reviewed in [9,10]), because the stem-mining larvae avoid direct contact with the pesticides [9]. To complement endophyte-based plant resistance [11,12], the solitary wasp *Microctonus hyperodae* Loan (Hymenoptera: Braconidae) was released for biological control of ASW in 1992. Within three years of its release, parasitism of ASW by *M. hyperodae* had reached 90% [13], reducing or eliminating damage to pasture [13,14,15].

Although ASW was initially suppressed by *M. hyperodae*, this began to fail after about 14 generations [16,17,18]. Loss of efficacy may be the result of adaptation in weevil populations resulting from selection pressure by the parasitoid [17,18]. Because ASW reproduces sexually, ASW populations may have greater capacity to evolve than populations of *M. hyperodae*, which reproduces parthenogenetically. Empirical modelling of the ASW–*M. hyperodae* interaction has indicated that resistance is inevitable when hosts have more genetic variation than their predators [19]. Despite this theoretical pathway for resistance, other examples of evolution of resistance to classical biological control have not been reported [20].

Population-level studies of genetic variation in host and parasitoid are required to explain the evolution of resistance in this case. We address this with a genotyping-by-sequencing study of a geographical survey of 10 Argentine stem weevil populations collected from across the North and South Islands of New Zealand. Our experiments revealed a repetitive genome with high heterozygosity and a high proportion of unstructured variation across populations. This is consistent with large effective population size and gene flow between populations. Genetic variation was found along a latitudinal cline, and was associated with signatures of selection in regions of the genome, indicating a level of local adaptation within populations, but at the resolution of this study we found no evidence of genetic adaptation in parasitised weevils compared to parasitoid-free weevils. Our results showed that the amount of genetic variation in New Zealand populations of ASW is far greater than detected by traditional molecular markers [21,22], implying that ASW populations have evolved resistance via weak selection acting on variants of minor effect that existed before the introduction of *M. hyperodae*.

## 2. Materials and Methods

### 2.1. Weevil Sampling

We collected ASW samples from commercially-farmed ryegrass (*Lolium perenne* L.)/white clover (*Trifolium repens* L.) pastures, at 10 sites across New Zealand, using a suction device to collect ground litter (Table 1). Weevils were extracted from the litter in the laboratory. The locations sampled are illustrated in Figure 1. The map was plotted with the ggmap package for ggplot2 [23], using map tiles by Stamen Design under CC BY 3.0, with data by OpenStreetMap under ODbL.

For the comparison between parasitised and unparasitised weevils, samples were collected from ryegrass/clover pasture at Ruakura and Lincoln (as in Table 1) in August 2017. These samples were dissected as described by Goldson and Emberson [24] to determine whether they were parasitised. After dissection, heads were removed and used for genotyping.

### 2.2. Genome Assembly

To produce the short read dataset, an Illumina TruSeq PCR-free 350 bp insert library was generated from DNA extracted from a single, male Argentine stem weevil collected from endophyte-free hybrid ryegrass (*L. perenne* × *Lolium multiflorum*) at Lincoln, New Zealand. Library preparation and sequencing were performed by Macrogen Inc. (Seoul, Korea). A total of 158 Gb of 100 b and 150 b paired-end reads were generated from the TruSeq PCR-free library. After removing common sequencing contaminants and trimming adaptor sequences using BBTools [25], the short-read-only genome was assembled with Meraculous 2.2.6 [26,27,28]. Reproducible code for assembling the short-read dataset and assessing the assemblies is hosted at github.com/tomharrop/asw-nopcr.

To produce long reads from a single individual, we produced high molecular weight DNA from a single, male ASW collected from Ruakura, New Zealand, using a modified QIAGEN Genomic-tip 20/G extraction protocol [29]. We amplified the DNA using Φ29 multiple displacement amplification (QIAGEN REPLI-g Midi Kit) and debranched the amplified DNA using T7 Endonuclease I (New England Biolabs) according to the Oxford Nanopore Technologies Premium whole genome amplification protocol version WGA_kit9_v1. Debranching reduced the raw read *N*_50_ length to 9.0 kb. Amplified DNA was sequenced on 6 R9.4.1 flowcells using a MinION Mk1B sequencer (Oxford Nanopore Technologies). We also extracted high molecular weight DNA from three pools, each of 20 unsexed individuals collected from Ruakura, New Zealand. We sequenced this pooled DNA on 5 R9.4.1 flowcells, following the Genomic DNA by Ligation protocol (SQK-LSK109; Oxford Nanopore Technologies). We basecalled raw Nanopore data with Guppy 3.4.1 (Oxford Nanopore Technologies). We removed adaptor sequences from the long reads with Porechop 0.2.4 (github.com/rrwick/Porechop) and assembled with Flye 2.6 [30].

All genome assemblies were assessed by size and contiguity statistics and BUSCO analysis [31]. Redundant contigs were removed from the combined, long read assembly with Purge Haplotigs 0b9afdf [32] using a low, mid and high cutoff of 60, 120 and 190, respectively.

We were not able to estimate repeat content in the full genomes, because RepeatModeler 2.0.1 [33] identified >500 M High-scoring Segment Pairs (HSPs) and did not finish after running for 6 weeks with ~200 GB of physical RAM (results not shown). We estimated repeat content by subsetting the assemblies using the leave-one-out alignment method implemented in Funannotate clean 1.7.4 [34]. We then used RepeatModeler 2.0.1 and RepeatClassifier 2.0.1 [33] and RepeatMasker 4.1.0 [35] from the Dfam TE Tools Container v1.1 (github.com/Dfam-consortium/TETools) to estimate the repeat content of the subset assemblies. We identified less than 1 M HSPs in the subset assemblies, indicating that the repeat content of the subset assemblies is an underestimate of the repeat content in the full assemblies.

Reproducible code for assembling and assessing the long-read ASW genomes is hosted at github.com/TomHarrop/asw-flye-withpool.

We annotated the final, draft assembly with Funannotate 1.7.4 [34], using five RNA sequencing libraries generated from abdomens and heads of unparasitised adult ASW collected from Ruakura. Reproducible code for annotating the draft ASW genome is hosted at github.com/TomHarrop/asw-annotate.

### 2.3. Reduced-Representation Genome Sequencing, Processing and Analysis

DNA extraction and double digest RADseq (genotyping-by-sequencing, GBS; [36]) were performed by AgResearch, Invermay, New Zealand. DNA was extracted from individual weevil heads using the ZR-96 Tissue & Insect DNA Kit (Zymo Research, CA, USA). The DNA was digested with *Ape*KI and *Msp*I and barcoded based on the Elshire method [37] with modifications [38]. Pooled libraries were size selected on a BluePippin (Sage Science, MA, USA) with a window size of 150–500 bp. 100 nt single-end reads were generated from libraries an Illumina HiSeq 2500 instrument.

We used a strict processing pipeline to prepare the raw GBS reads for locus assembly. Samples were demultiplexed with zero allowed barcode mismatches to 91–93 b reads, depending on barcode length. Reads were trimmed by searching for adaptors with a minimum match of 11 b. Reads shorter than 80 b after trimming were discarded. All remaining reads were truncated to 80 b to account for unmatched adaptor sequence < 11 b that may have been present at the end of reads. To remove overamplified samples, we calculated the GC content for each library and discarded samples with median read GC > 45%. We assembled loci against our draft genome using gstacks 2.53 [39].

For analysis, we used BCFtools 1.10 to remove sites with more than 2 alleles, minor allele frequency <0.05, or missing genotypes in more than 20% of individuals. After filtering loci, we also removed individuals that had missing genotypes at more than 20% of loci. We ran the Stacks 2.53 populations module [39] to calculate inbreeding (*F*) and heterozygosity statistics. We used PLINK 1.9 [40] to prune sites in linkage disequilibrium for principal components analysis and discriminant analysis of principal components with the adegenet 2.1.2 package for R [41,42], using the first four principal components. We used PGDSpider 2.1.1.5 [43] to convert the un-pruned dataset for detection of loci under selection with BayeScan 2.1 [44]. We analysed cross-population extended haplotype homozygosity with the R package rehh 3.1.0 [45]. For demographic analysis, we converted the pruned dataset to minor allele (folded) site frequency spectra using easySFS commit c2b26c5 from github.com/isaacovercast/easySFS. We estimated likelihood for each demographic model ten times using fastsimcoal2 2.6 [46] with 1 million simulations and 60 optimisation cycles per run. We compared model runs using delta likelihood (maximum observed likelihood - maximum estimated likelihood) and Akaike information criteria [47].

All the code we used to process the raw reads, assemble loci and run downstream analyses is hosted at github.com/TomHarrop/stacks-asw, including the parameters and software containers for each step.

### 2.4. Reproducibility and Data Availability

Raw sequence data for the ASW genome assembly and annotation and raw GBS reads are hosted at the National Center for Biotechnology Information Sequence Read Archive (NCBI SRA) under accession PRJNA640511. We used Snakemake [48] to arrange analysis steps into workflows and monitor dependencies, and Singularity [49] to capture the computing environment. Using the code repositories listed in each methods section, the final results can be reproduced from the raw data with a single command using Snakemake and Singularity. The source for this manuscript is hosted at github.com/TomHarrop/asw-gbs-genome-paper.

## 3. Results

### 3.1. The Argentine Stem Weevil Genome Is Repetitive and Polymorphic

To construct a reference for genotyping populations of Argentine stem weevils, we produced a draft assembly of the ASW genome. We initially attempted assembly from a single individual using PCR-free, short read sequencing. This resulted in a fragmented assembly with low BUSCO scores (Table 2). *k*-mer analysis on the raw short reads suggested 2.1 polymorphisms per 100 bp and a genomic repeat content of 28–48% in the individual we sequenced (Appendix A). We then attempted to produce a long-read genome assembly using whole-genome amplification (WGA) of high molecular weight (HMW) DNA from a single individual, followed by sequencing on the Oxford Nanopore Technologies (ONT) MinION sequencer. We produced 29.8 Gb of quality-filtered reads with an *N*_50_ length of 9.0 kb. Assembling the single individual, long read genome resulted in improved contiguity and BUSCO scores compared to the short-read assembly (Table 2). Consistent with the raw short read data, the single individual, long read genome was at least 70% repetitive (Table 2). To improve assembly across long repeats, we produced a second ONT dataset with longer reads from HMW DNA from three pools of 20 individuals each, without amplification. Sequencing these samples on the MinION sequencer produced a total of 12.0 Gb of quality-filtered reads with an *N*_50_ length of 19.5 kb. Assembling the longer reads generated from the pooled sample alone resulted in a more contiguous genome, but with lower BUSCO scores (Table 2). We constructed a combined, long-read genome using the pooled, long-read dataset for contig construction, and the single-individual, long-read dataset for assembly polishing. This improved the BUSCO scores, but produced a large number of redundant contigs (Table 2), presumably because of the high rate of heterozygosity in the pooled, long-read dataset. We then used the PCR-free, short read sequencing data from a single individual with the Purge Haplotigs pipeline to remove redundant contigs from the combined long read assembly [32]. This resulted in a final draft assembly of 1.1 Gb with an *N*_50_ length of 122.3 kb and a BUSCO completeness of 83.9%. The final draft assembly had a repeat content of at least 70% (Table 2; Appendix A), with a maximum repeat size of 30.4 kb and a repeat *N*_50_ length of 494 bp. The majority of the repeats were unclassified when compared against the Dfam 3.1 database [50], with 9.2% of the genome detected as retroelements and 7.5% as DNA transposons (Appendix A) The non-repetitive regions had an *N*_50_ length of 1066 bp.

### 3.2. Genetic Variation Is Associated with Geography in NZ Populations of Argentine Stem Weevil

To measure genetic variation in invasive New Zealand populations of ASW, we collected individuals from 10 sites across the North and South Islands of New Zealand (Figure 1A). We genotyped 183 individuals with a modified genotyping-by-sequencing (GBS) protocol [37,38]. After strict trimming and filtering of the raw GBS data, we mapped reads from each individual against our draft genome and used gstacks to assemble loci [39]. For analysis, we removed loci with more than two alleles, minor allele frequency less than 0.05, or missing genotypes in more than 20% of individuals. We also removed individuals missing genotypes at more than 20% of loci. The filtered dataset comprised 7–15 individuals per location (total 116), genotyped at 52,051 biallelic SNPs. The mean observed heterozygosity ranged from 0.18–0.21 across populations (Figure 1B), and pairwise *F*_ST_ values between populations ranged from 0.024–0.051 (Figure 1C). For principal components analysis (PCA), we pruned the dataset to 18,715 biallelic SNPs that were not in linkage disequilibrium, using a correlation threshold of 0.1. PCA of genotypes at these sites revealed overlapping populations of ASW, with 9.2% of total variance explained by the first two components (Figure 1D). These populations of ASW are highly heterozygous, but the low proportion of total variance explained by the major principal components suggests that variation is not highly structured between populations. This is consistent with a large effective population size and gene flow between populations. To find variance between populations, we used discriminant analysis of principal components (DAPC) on the same set of pruned SNPs [41]. The major linear discriminant, which explains 96.7% of between-population variation, separates populations from North and South of the Main Divide (Figure 1E), although we found evidence of mixing in all populations except Lincoln (Figure 1F). Although the PCA suggests that the majority of the total variance is not structured, the DAPC indicates a degree of genetic isolation between populations from North and South of the Main Divide. This suggests that the Main Divide, which runs along the Southern Alps and divides the South Island, is the main geographic barrier to ASW populations in New Zealand.

### 3.3. Genetic Variation Is not Associated with Parasitism by M. hyperodae

To detect large-effect variants associated with susceptibility to parasitism by *M. hyperodae*, we genotyped weevils that had also been tested for the presence of a parasitoid larva. We used a total of 200 individuals, collected from Lincoln and Ruakura (Table 3), because of the extent of historical declines in parasitism rates recorded at these locations [18]. The weevils were examined for a parasitoid larva and genotyped at the same loci used for the geographical diversity survey. After filtering and pruning sites in linkage disequilibrium, we used 19,482 SNPs for PCA and DAPC in 95 parasitised inviduals and 84 individuals where a parasitoid was not detected (Table 3). We did not detect any genetic differentiation associated with the presence of a parasitoid, either within populations or between populations, or any evidence of skewed allele frequencies in these groups using BayeScan (lowest *Q*-value 0.97).

### 3.4. Genetic Differentiation between ASW Populations North and South of the Main Divide

Although we did not detect variation associated with presence of a parasitoid, parasitism rates vary across sites in NZ [18]. Regional genetic differences could be related to selection acting on different loci North and South of the Main Divide and/or genetic drift acting on isolated populations. To investigate the genetic differentiation between regions, we grouped individuals that were collected from North and South of the Main Divide (Figure 1). The two groups had a mean *F*_ST_ of 0.013. We detected 47 SNPs with skewed allele frequencies across 24 contigs in the draft genome with BayeScan (Figure 2). The contigs containing these SNPs had a total of 3–36 SNPs, and all 47 of the detected SNPs had positive α values, suggesting diversifying selection (Appendix A). The SNPs identified by BayeScan were an average of 11.5 kb from the nearest genes. None of the closest genes were homologous to genes with characterized functions in insects. Using an additional method, 3 SNPs on another contig had outlying cross-population extended haplotype homozygosity (XPEHH) scores (Appendix A; [51,52]). No common regions were identified by both methods.

### 3.5. Separate Incursions of ASW into New Zealand

The genetic differentiation between weevils from North and South of the Main Divide suggests the possibility of either multiple routes of entry, or incursion via a single route of entry followed by isolation and diversification. The level of heterozygosity we measured across populations also suggests that incursions were large and/or repeated. To test these different possibilities, we simulated site frequency spectra (SFS) under 10 different models of demographic history and compared them to the observed SFS. Our models covered single and multiple introductions, with either moderate or strong reduction in effective population size during introduction, and multiple introductions from different source populations (Figure 3A). The models that best matched the observed SFS had the North and South populations separated before bottlenecks (Figure 3A, models *ii*, *iii* and *v*), and support for these models was better when migration between populations was included. The model with the lowest mean delta likelihood supports separate routes of entry into New Zealand, with a bottleneck in each population prior to entry and migration between the North and South populations (Figure 3B).

## 4. Discussion

The purpose of this work was to investigate genetic variation in New Zealand populations of ASW and its possible relationship to resistance to *M. hyperodae*. Previous reports using randomly amplified polymorphic DNA (RAPD) markers and *cytochrome C oxidase subunit I* (*COI*) sequencing suggested a high degree of genetic similarity and identified a single *COI* haplotype in New Zealand populations [21,22]. In contrast, our results from a genome-wide genotype-by-sequencing (GBS) approach reveal a high level of genetic diversity within and between populations. We suggest that this standing variation provides an evolutionary advantage to ASW populations in comparison to their biocontrol agent, *M. hyperodae*. We expect variation to be limited by asexual reproduction in *M. hyperodae* (e.g., [19]). This lack of variation, and the inability of *M. hyperodae* to switch hosts [53], would limit the capacity of *M. hyperodae* to co-evolve with ASW. This indicates that genetic variation in both host and biocontrol agent need to be monitored with high-resolution genotyping to maintain success of biological control. More work will be required to describe the genetic mechanism of resistance and its prevalence in weevil populations, and to measure the amount of variation and population structure in *M. hyperodae*.

ASW was thought to have arrived in New Zealand in the early 20th century, probably via trade in pasture seeds or hay used for feed during stock transit [54]. The earlier reports of low genetic diversity, based on traditional molecular markers [21,22], suggested a limited incursion followed by dispersal and expansion. Our results provide three main pieces of evidence to update the proposed history of ASW incursions in New Zealand. The high heterozygosity across populations could be explained by a large initial incursion, repeated introductions, and/or an unusually high mutation rate. The genetic differentiation between populations from North and South of the Main Divide points to low migration rates between these regions. Our demographic modelling suggests that the populations expanded to their current effective sizes after already being separated into North and South populations. The most likely scenario is separate introductions from the same source population to North and South of the Main Divide, with some migration between the two populations. The power to resolve the possible evolutionary histories that led to the current population structure of New Zealand weevils was provided by the increased resolution of genome-wide genotyping.

Despite the increased resolution of GBS compared to traditional markers, we did not detect regions of the genome associated with parasitism by *M. hyperodae*. Possible reasons for this include one or more of the following: *i*. resistance to biocontrol may not be genetic; *ii*. resistance may be encoded by part of the genome not captured in our assembly; *iii*. microscopic detection of the parasitoid may not be a strong enough phenotype to separate resistant and susceptible individuals, because individuals without a detectable parasitoid are not necessarily resistant, e.g., if they had not been exposed to the parasitoid before collection from the field; or *iv*. resistance is encoded by multiple regions of small effect, which we were unable to detect in our study. In model organisms, adaptive evolution in response to selective agents acting within the survivability distribution of a population takes the form of polygenic responses on standing variation [55,56]. The highest reported parasitism rate of ASW by *M. hyperodae* is 90% [13], implying that some individuals in a population survive predation. In other words, selection by *M. hyperodae* acts within the survivability distribution of ASW populations. Because we detected a large amount of standing variation in our survey of ASW populations, which may encode phenotypic variation in parasitism survivability, we suggest that a polygenic response is the most probable scenario. The number of markers yielded by legacy genotyping-by-sequencing approaches provides low power to detect polygenic responses resulting from weak selection on standing variation. Higher-resolution, genome-wide association studies using whole-genome resequencing with more individuals and a stronger resistance phenotype may allow detection of regions of the genome associated with resistance of the weevils to biocontrol.

Two draft weevil (Coleoptera: Curculionidae) genomes constructed from short reads have been deposited in the NCBI Genome database. The coffee berry borer, *Hypothenemus hampei*, has a draft genome size of 163 MB [57], and the mountain pine beetle, *Dendroctonus ponderosae*, has a draft genome size of 202 MB in males and 213 MB in females [58]. Draft genomes that incorporate long reads have been deposited for the red palm weevil (*Rhynchophorus ferrugineus*; GCA_012979105.1) and the rice weevil (*Sitophilus oryzae*; GCF_002938485.1). These assemblies are 782 MB and 771 MB, respectively. Assemblies using long reads capture more of the genome, presumably because larger repeat regions can be assembled. Our ASW genome of 1.1 GB is larger than other available weevil genomes, and has a high proportion of repetitive sequences. The contiguity statistics and BUSCO scores indicate draft quality, and we expect gaps in the assembly at larger repeat regions that were not sufficiently covered by long reads. Our attempt at short-read assembly of the Argentine stem weevil genome was not effective because of the extreme repeat content. The heterozygosity in weevil populations and lack of an inbred, laboratory strain made pooling individuals for sequencing undesirable. This is highlighed by the number of multiple-copy genes in the combined, long read assembly. Our strategy to assemble the ASW genome included contig construction with the longest reads, followed by assembly polishing with long reads from a single individual, and then redundant contig removal with PCR-free short reads from another single individual. This allowed us to optimise the contiguity and completeness of the genome whilst managing the number of redundant contigs (Table 2).

## 5. Conclusions

Our results show that New Zealand populations of ASW have a large amount of heterozygosity, and we suggest that this allowed them to evolve resistance to their biological control agent. This highlights the need for monitoring biological control systems by genome-wide genotyping.

## Figures and Tables

**Figure 1 insects-11-00441-f001:**
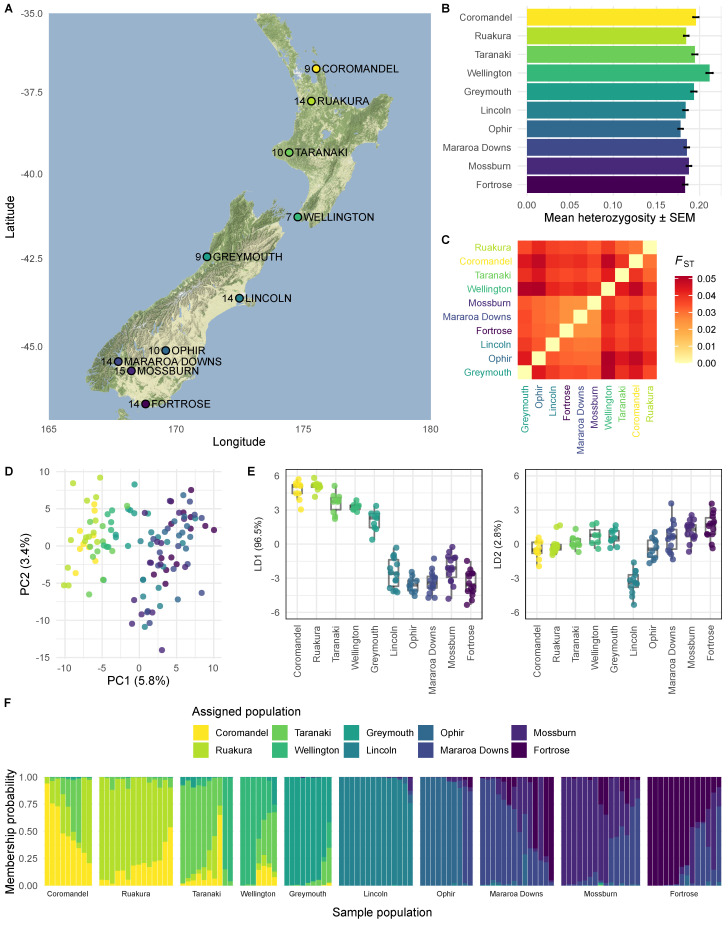
Genetic diversity in NZ populations of Argentine stem weevil. (**A**) Weevil sampling locations. We collected Argentine stem weevils from 4 locations in the North Island and 6 locations in the South Island of New Zealand. Greymouth is in the South Island, but North of the Main Divide, which runs along the Southern Alps and partitions the South Island. The number of weevils genotyped (after filtering) from each location is shown on the map. Map tiles by Stamen Design under CC BY 3.0, with data by OpenStreetMap under ODbL. (**B**) Mean observed heterozygosity for each population. (**C**) Pairwise *F*_ST_ values between populations. (**D**) Principal components analysis (PCA) describing total variability and (**E**) discriminant analysis of principle components (DAPC) describing between-population variability of 116 individuals genotyped at 18,715 biallelic sites. In the PCA (**D**), populations overlap on the first two principal components (PC1 and PC2), but weevils sampled from higher latitudes have lower scores on PC1. PC1 and PC2 together explain 9.2% of variance in the dataset, indicating a high level of unstructured genetic variation in weevil populations. In the DAPC (**E**), the major linear discriminant (LD1) explains 96.7% of between-group variability. LD1 splits individuals from North and South of the Main Divide. LD2 separates Lincoln individuals from other individuals. (**F**) Posterior probability of group assignment for each individual. All populations contain individuals with high posterior probabilities of membership to other populations. This is consistent with gene flow between populations. We did not detect evidence of gene flow between populations from North and South of the Main Divide. Individuals sampled from Lincoln had the lowest posterior probabilities of membership to other populations.

**Figure 2 insects-11-00441-f002:**
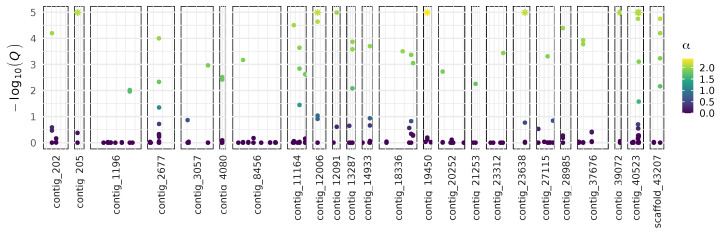
Distribution of SNPs that are associated with altered allele frequencies between populations from North and South of the Main Divide over 24 contigs. 47 SNPs have altered allele frequencies, using the arbitrary Q-value cutoff of 0.01. α: BayeScan’s locus-specific component of F_ST_ coefficient [44]. Positive values suggest diversifying selection.

**Figure 3 insects-11-00441-f003:**
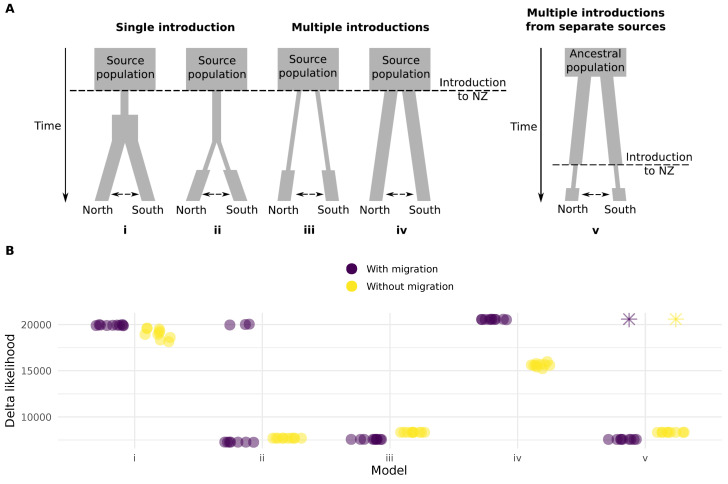
Models used to simulate site frequency spectra (SFS) for comparison against the observed SFS. (**A**) Models i and ii represent introduction via a single route. In model i the population contracts on introduction and then begins expanding before dispersal. In model ii, dispersal occurs while the effective population size remains small. Models iii and iv represent multiple routes of entry, with or without reduction in population size compared to the current populations in New Zealand. Model v represents introduction to different sites from source populations that were isolated prior to introduction. All models were tested with and without migration between Northern and Southern populations. (**B**) Likelihood estimations from ten runs of each model. We used 1M simulations and 60 optimisation cycles per run [46]. Runs that ended with a non-finite delta likehood after 1M simulations are shown as an asterisk. The models that include a bottleneck after separation of the two populations had the lowest delta likelihoods, and model iii (with migration) had the lowest mean delta likelihood across runs.

**Table 1 insects-11-00441-t001:** Weevil collection locations (see also Figure 1).

Location	GPS Co-Ordinates (lat, lon)	Date Collected	Number Genotyped
Coromandel	−37.20194, 175.59417	June 2015	16
Ruakura	−37.76750, 175.32361	June 2015	16
Taranaki	−39.61500, 174.30278	July 2015	15
Wellington	−41.13647, 175.35163	July 2015	16
Greymouth	−42.89506, 172.71926	September 2016	16
Lincoln	−43.64397, 172.44292	July 2014	15
Ophir	−45.10955, 169.58753	August 2017	15
Mararoa Downs	−45.50672, 167.97596	May 2016	16
Mossburn	−45.66966, 168.23884	January 2016	16
Fortrose	−46.57064, 168.79993	November 2016	16

**Table 2 insects-11-00441-t002:** Assembly statistics for the final draft genome and intermediate assemblies. 1. We estimated repeat content from a subset of contigs in the assembly (see Methods). n.d.: not determined.

	Short Read	Single Individual, Long Read	Pooled, Long Read	Combined, Long Read	Final Draft
Assembly length (Gb)	1.3	1.2	1.2	1.7	1.1
*N* _50_	53,046	4523	2958	5281	2681
*N*_50_ length (kb)	7.1	74.4	112.6	86.4	122.3
Gaps (%)	3.5	0	0	0	0
GC content (%)	30.6	31.3	31.4	31.4	31.3
Complete, single-copy BUSCOs (%)	32.7	72.2	71	69.2	78.8
Complete, multiple-copy BUSCOs (%)	17.2	7.5	5.9	17.4	5.1
Minimum ^1^ repeat content (%)	n.d.	71	71.4	71.4	71.3

**Table 3 insects-11-00441-t003:** Number of parasitised and unparasitised weevils from Ruakura and Lincoln.

Location	Parasitoid	Number Genotyped	Number after Filtering
Ruakura	Present	50	46
Ruakura	Not detected	50	40
Lincoln	Present	50	49
Lincoln	Not detected	50	44

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
