# Peer review of "Genetic Diversity in Invasive Populations of Argentine Stem Weevil Associated with Adaptation to Biocontrol"

_insects, 2020, doi:10.3390/insects11070441_

Round 1
Reviewer 1 Report
The manuscript entitled, “Genetic diversity in invasive populations of Argentine stem weevil allows adaptation to biocontrol”, adds an additional piece to a fascinating story and would be of interest to the readership of Insects. Previous work had demonstrated that the Argentine stem weevil (Listronotus bonariensis, ASW), a significant economic pest of pasture grass in New Zealand, was initially controlled by an introduced parasitic wasp (Microtonus hyperodae). However, ASW populations were only suppressed for about 14 generations, after which parasitism rates declined. Previously published analytical models supported the hypothesis that the level of genetic variation present in ASW – which reproduces sexually – relative to the parasitoid – which reproduces asexually – could have facilitated ASW’s rapid evolution to and escape from parasitism pressure. Therefore, the authors in this study characterized the level of genomic variation present in the ASW genome and used molecular markers to compare the genetic material of parasitized and healthy ASW seeking evidence of adaptive evolution. The manuscript is well-written and of sound experimental methodology. I have only minor comments and suggestions, mainly involving possible additions to the discussion.
TITLE
The title might be a bit of a stretch; authors were able to document high genetic diversity across ASW populations but are unable to provide clear link to the adaptation to biocontrol. Would the authors consider hedging (“could allow adaptation to biocontrol” or something more along those lines)?
METHODS
Lines 76-77, as a reader familiar with your previous work on the subject, I wondered why not compare ASW collected in Lolium perenne/clover pastures with ASW collected from L. multiflorum pastures, since the latter hosted weevils with higher parasitism rates (Tomasetto et al 2017)? It would be very interesting if host-associated genetic divergence were related to parasitism survivorship.
RESULTS
Lines 199-201, this information is also present in the methods section and could be omitted.
Figure 1, I found this figure especially helpful for the interpretation of results and appreciated that the authors described the location of the Main Divide relative to collection sites.
DISCUSSION
Lines 325-326, it is somewhat unclear what the authors mean by point iii. Is it that microscopic detection of parasitism might not be a good indicator of ultimate ASW mortality/parasitism success?
Lines 348-350, if concerned about gaps in the reference genome assembly relative to the usage of SNPs to infer genetic divergence, the authors could additionally run a de novo analysis and compare to identify whether there are outlier loci that don’t show up in the reference genome and to see how many loci might be missing from it (Rochette & Catchen 2017). However, the BUSCO score seemed to be of fairly good quality (~84%). So, comparison with a de novo approach could be informative but might not be necessary.
There were a few additional discussion topics I found myself wanting to read about relative to these results. One was that including a discussion of adaptation rates within sexual vs. asexual populations would be appropriate for context given the story and conclusion. Another interesting discussion might involve comparisons with other biocontrol successes and failures (although as the authors stated in the introduction, this remains the sole record of pest out-evolving its biocontrol agent), especially relative to genomic diversity/sexual vs. asexual reproductive modes. Lastly, the parasitoid would have undergone similarly high selection pressure. Do the authors think it possible that the reduced parasitism rates in ASW reflect host switching by the parasitoid? Is it a specialist or can it parasitize more than one host species?
Author Response
Reviewer's comments in italics, with our response below.
TITLE
The title might be a bit of a stretch; authors were able to document high genetic diversity across ASW populations but are unable to provide clear link to the adaptation to biocontrol. Would the authors consider hedging (“could allow adaptation to biocontrol” or something more along those lines)?
We have changed the title to “Genetic diversity in invasive populations of Argentine stem weevil is associated with adaptation to biocontrol”.
METHODS
Lines 76-77, as a reader familiar with your previous work on the subject, I wondered why not compare ASW collected in Lolium perenne/clover pastures with ASW collected from L. multiflorum pastures, since the latter hosted weevils with higher parasitism rates (Tomasetto et al 2017)? It would be very interesting if host-associated genetic divergence were related to parasitism survivorship.
There are two reasons we chose these sites:
-
the weevils are mobile and we would expect the genotypes in neighbouring L. perenne and L. multiflorum stands to be similar (see 10.1111/j.1365-3032.1981.tb00272.x, 10.1017/S0007485399000553 and 10.1093/ee/18.6.996);
-
L. multiflorum pastures are ephemeral, usually persisting for less than a year, after which they are recultivated (often with L. perenne).
RESULTS
Lines 199-201, this information is also present in the methods section and could be omitted.
We’ve left this in, because we think it’s important for readers to understand the filtering we did before analysis without having to read the methods section.
Figure 1, I found this figure especially helpful for the interpretation of results and appreciated that the authors described the location of the Main Divide relative to collection sites.
We’re glad to hear this figure and the caption helped understanding of the paper.
DISCUSSION
Lines 325-326, it is somewhat unclear what the authors mean by point iii. Is it that microscopic detection of parasitism might not be a good indicator of ultimate ASW mortality/parasitism success?
Reviewer #2 also mentioned point iii from this paragraph. We’ve made this point clearer by changing the wording as follows: “iii. microscopic detection of the parasitoid may not be a strong enough phenotype to separate resistant and susceptible individuals, because individuals without a detectable parasitoid are not necessarily resistant, e.g. if they had not been exposed to the parasitoid before collection from the field;”
Lines 348-350, if concerned about gaps in the reference genome assembly relative to the usage of SNPs to infer genetic divergence, the authors could additionally run a de novo analysis and compare to identify whether there are outlier loci that don’t show up in the reference genome and to see how many loci might be missing from it (Rochette & Catchen 2017). However, the BUSCO score seemed to be of fairly good quality (~84%). So, comparison with a de novo approach could be informative but might not be necessary.
This is a good point. We did the same analyses without a reference genome (i.e. using the ‘de novo’ Stacks pipeline) and got the same results for the geographical survey: there is a high amount of heterozygosity, most of the variation is unstructured, a there is a genetic cline associated with latitude. Using the de novo approach, we also failed to detect skewed allele frequencies between parasitised an unparasitised individuals. We decided not to include these results in the manuscript, because they repeat the results we got from the reference-guided approach but lack information about linkage and physical proximity between markers.
There were a few additional discussion topics I found myself wanting to read about relative to these results. One was that including a discussion of adaptation rates within sexual vs. asexual populations would be appropriate for context given the story and conclusion. Another interesting discussion might involve comparisons with other biocontrol successes and failures (although as the authors stated in the introduction, this remains the sole record of pest out-evolving its biocontrol agent), especially relative to genomic diversity/sexual vs. asexual reproductive modes.
There is no other documented classical biological control failure, let alone a study that covers the effect of a parthenogenetic parasitoid active against a sexually-reproducing host. However, the the work of Casanovas et al. (2018; 10.1371/journal.pone.0207610) is pertinent inasmuch that their modelling work was based on parameters obtained from other extensive historical field and laboratory biology and ecology datasets.
Lastly, the parasitoid would have undergone similarly high selection pressure. Do the authors think it possible that the reduced parasitism rates in ASW reflect host switching by the parasitoid? Is it a specialist or can it parasitize more than one host species?
We have added discussion of host specificity to the discussion as follows: “This lack of variation, and the inability of M. hyperodae to switch hosts, would limit the capacity of M. hyperodae to co-evolve with ASW”. This includes a citation to the following paper: 10.1007/BF02373121.
Reviewer 2 Report
The work by Harrop et al. explores the genetic diversity in the Argentine stem weevil through genomic approaches and seeks the causes of resistance to biological control by means of a solitary wasp. First, the authors got the genome assembly, and later they genotyped a sample collected in both New Zeland´s islands and used the genome to align this dataset. The work is well written, easy to follow, the genome assembling is accurately made (different approaches have been used to get a good one) and the parameters of the genotyping by sequencing approach seem appropriate to me (MAF, missing genotypes and so on). Sample sizes in all approaches look quite good. Results are quite interesting and well discussed. Although I recommend acceptance of this manuscript, I suggest some minor concerns to get solved before publication.
MINOR CONCERNS:
Line 19: “but this depends on the agent’s ability to adapt to evolution in the host”. This sentence sounds really funny. What does it mean “to adapt to evolution”? This is nonsense to me. I suggest something like “elude the host resistance resulting from natural selection//evolution”, or something like that.
Line 19: “unequal genetic variation”. This also sounds quite funny. I suggest “different levels of genetic variation//diversity”
Line 23: delete the coma after “pasture pest,”.
Line 23: I think the scientific name is very important to be between parentheses.
Line 76: “We collected regional ASW samples”. I think you better sample individuals at a regional scale.
Line 83: “dissected as per Goldson and Emberson ”, replace by “dissected as Goldson and Emberson”
Line 138: Replace “bcftools” by “BCFtools”. By the way, many programs should be capitalized (e.g. snakemake, miraculous, funannotate clean)
In Results, it might be useful add a column to Table 1 with sample size per population.
Lines 206-208: Authors think that PCA does not show differentiation between populations from north and south of the country. I suggest reanalyzing along the PC1: I can see two clouds (although there is a bit of overlap). This can also be seen along the PC2. A bit of differentiation can be inferred from this analysis.
Lines 215-217: I disagree. PCA shows differentiation. Maybe an AMOVA will show differentiation between the north and south islands, if two clusters are define (south and north), and still most of the variation will be seen within populations. But that small percentage of variation will be significant.
Lines 228-230: Add “data not shown”.
Lines 317-319: Authors stated that “The most likely scenario is separate introductions from the same source population to North and South of the Main Divide, with some migration between the two populations”. However, in lines 279-281 they suggested that model v) was the preferred one. Thus, as I understand, by looking model v) in Figure 3, I think they should state “The most likely scenario is separate introductions from different populations to North and South of the Main Divide, with some migration between the two populations”. Maybe I am wrong, but in this case, the authors should clarify these sentences because there is some kind of inconsistency.
Regarding the absence of outlier loci in parasitized vs. Non-parasitized weevils, and I am asking this because I know nothing about the mechanism by which the Argentine stem weevil resists parasitoid infection, would it be possible that those non-parasitized weevils were collected before having the opportunity of being parasitized? I mean, if weevils kill the egg parasitoid when it injected into the adult (if they are parasitoids in the adult state as occur in some weevils, or in the egg, I do not know what happens in the ASW), outlier loci should be studied in surviving weevils, not in non-parasitized. Maybe the ones non-parasitized are free from parasitoids just by chance. I repeat, I do know nothing about the mechanism of resistance in the ASW.
Lines 346-347: “Assemblies using long reads capture more of the genome, presumably because larger repeat regions can be assembled”. Yes, but they are also prone to more errors!
Author Response
Reviewer's comments in italics, with our response below.
MINOR CONCERNS:
Line 19: “but this depends on the agent’s ability to adapt to evolution in the host”. This sentence sounds really funny. What does it mean “to adapt to evolution”? This is nonsense to me. I suggest something like “elude the host resistance resulting from natural selection//evolution”, or something like that.
We’ve changed this to read “this depends on the agent’s capacity to co-evolve with the host”. We haven’t used the phrase “elude the host resistance resulting from natural selection//evolution”, because we’re not necessarily talking about resistance here.
Line 19: “unequal genetic variation”. This also sounds quite funny. I suggest “different levels of genetic variation//diversity”
We have changed this to read “Theoretical studies have shown that different levels of genetic variation between the host and the control agent will lead to rapid evolution of resistance in the host”.
Line 23: delete the coma after “pasture pest,”.
Line 23: I think the scientific name is very important to be between parentheses.
We’ve haven’t made these changes because we think the original reads better, but we’re happy for the copy editor to change as necessary.
Line 76: “We collected regional ASW samples”. I think you better sample individuals at a regional scale.
We have changed this to read “We collected ASW samples from commercially-farmed ryegrass (Lolium perenne L.) / white clover (Trifolium repens L.) pastures, at 10 sites across New Zealand, using a suction device to collect ground litter (Table 1)”.
Line 83: “dissected as per Goldson and Emberson”, replace by “dissected as Goldson and Emberson”
We have changed this to read “These samples were dissected as described by Goldson and Emberson”.
Line 138: Replace “bcftools” by “BCFtools”. By the way, many programs should be capitalized (e.g. snakemake, miraculous, funannotate clean)
We’ve checked spelling of all the software we used and changed the ones we had capitalised incorrectly.
In Results, it might be useful add a column to Table 1 with sample size per population.
We’ve added the number of individuals genotyped to table 1, and clarified in the legend to figure 1 that the numbers on the map refer to the number of individuals used after filtering our genotyping results.
Lines 206-208: Authors think that PCA does not show differentiation between populations from north and south of the country. I suggest reanalyzing along the PC1: I can see two clouds (although there is a bit of overlap). This can also be seen along the PC2. A bit of differentiation can be inferred from this analysis.
Lines 215-217: I disagree. PCA shows differentiation. Maybe an AMOVA will show differentiation between the north and south islands, if two clusters are define (south and north), and still most of the variation will be seen within populations. But that small percentage of variation will be significant.
Lines 228-230: Add “data not shown”.
We agree with the reviewer’s interpretations of these data. These three comments seem to be related to a misunderstanding of the PCA results. The PCA is shown in Figure 1D, so it would be incorrect to label it with “data not shown”. We agree that there is some differentiation on PC1, which is why we wrote “In the PCA (D), populations overlap on the first two principal components (PC1 and PC2), but weevils sampled from higher latitudes have lower scores on PC1” in the figure legend. Separation on PC2 is more subtle, and possibly related to within-population variance. The method we used to analyse the PCA (i.e. DAPC) provides similar information to AMOVA, and we think our DAPC adequately describes the genetic differentiation between North and South (Figure 1E and 1F).
Lines 317-319: Authors stated that “The most likely scenario is separate introductions from the same source population to North and South of the Main Divide, with some migration between the two populations”. However, in lines 279-281 they suggested that model v) was the preferred one. Thus, as I understand, by looking model v) in Figure 3, I think they should state “The most likely scenario is separate introductions from different populations to North and South of the Main Divide, with some migration between the two populations”. Maybe I am wrong, but in this case, the authors should clarify these sentences because there is some kind of inconsistency.
Demographic models ii, iii and v all have the populations separated before the bottleneck. To make this clearer, we have changed this sentence to read “The models that best matched the observed SFS had the North and South populations separated before bottlenecks (Figure 3A, models ii, iii and v)”.
Regarding the absence of outlier loci in parasitized vs. Non-parasitized weevils, and I am asking this because I know nothing about the mechanism by which the Argentine stem weevil resists parasitoid infection, would it be possible that those non-parasitized weevils were collected before having the opportunity of being parasitized? I mean, if weevils kill the egg parasitoid when it injected into the adult (if they are parasitoids in the adult state as occur in some weevils, or in the egg, I do not know what happens in the ASW), outlier loci should be studied in surviving weevils, not in non-parasitized. Maybe the ones non-parasitized are free from parasitoids just by chance. I repeat, I do know nothing about the mechanism of resistance in the ASW.
This comment is correct. Non-parasitised weevils are not necessarily resistant, they may have escaped parasitism and/or exposure. We had intended to convey this with the sentence “iii. microscopic detection of the parasitoid may not be a strong enough phenotype to separate resistant and susceptible individuals”. We’ve clarified this by changing the wording as follows: “iii. microscopic detection of the parasitoid may not be a strong enough phenotype to separate resistant and susceptible individuals, because individuals without a detectable parasitoid are not necessarily resistant, e.g. if they had not been exposed to the parasitoid before collection from the field;”
Lines 346-347: “Assemblies using long reads capture more of the genome, presumably because larger repeat regions can be assembled”. Yes, but they are also prone to more errors!
The reviewer is correct about the increased error rate in raw long reads. We took advantage of recent advances in Nanopore basecalling (Guppy > 3) and assembly (Flye > 2.5), which allow long-read-only assemblies to approach the base-correctness of Illumina-only assemblies. We’ve added the following sentence to the methods section: “We basecalled raw Nanopore data with Guppy 3.4.1 (Oxford Nanopore Technologies)”. We did try ‘polishing’ the long-read-only assemblies with Illumina reads, but this actually resulted in worse BUSCO scores. We haven’t included these results in the manuscript.
Reviewer 3 Report
Dr. Harrop and collaborators propose a genetic study on the Argentine stem weevil in New Zealand. They obtained a draft genome for the species, study the geographic structuring of populations and the invasion history, and test for selection at the genome level associated to the distinction between North and South population and to observed pararitism by Microctonus parasitoid.
I think that, with some limitations, this is a good contribution, worth publishing in Insects.
Genome assembly, and the final quality of the draft, was greatly complicated by the repetitive structure of the genome and by the non availability of a reference inbred strain. Basing the sequencing on few individuals from the same sampling time/place may have been a good idea in the beginning. Nevertheless (to my understanding) all possible post hoc strategies to improve the genome were put forward.
Selection analysis did not produce significant results (i.e. the identification of regions/genes under specific selection) but the associated discussion, presenting different hypotheses for this observation, is correct.
The population genetic part provided the most interesting results of this study, leading to a description of the levels of heterozygosity, an overall picture of population structuring and an hypothesis for the colonization process.
In terms of the general structure of the manuscript, my main concern regards the possibility to provide a biological case example in support for the theoretical hypothesis that, in the competition between pararite and control agent, the one with the higher genetic variability will prevail. This is obviously presented as the driving rationale of the study, as it occupies the initial part of the abstract ad introduction and the conclusions. Nevertheless, in my view, this study cannot provide explicit support for the theory because there is no control, or baseline values, for variability and no data at all for the parasitoid. I would say that the observed high variability for the pest, the hypothesized low variability of the parasitoid and the observation that the parasitoid is loosing effectiveness after some generations are in line with the theory, but do not provide a nice and clean study case in its support. I would suggest presenting the study more simply as a genetic study of an important pest, not as a biological proof of the theory. The theory can obviously be introduced and discussed, but I do not see it as the main focus.
Do authors have clues about the nature of the repeated sequences?
Figures are complex and require extensive captions. Nevertheless captions now include quite some imformations that may be better placed in the methods or results sections.
The rationale behind testing for selection in parasitized/non parasitized pests is clear. Please clarify better the rationale for testing for selection between Northern and Southern populations once the former has proved ineffective. Are authors still looking for selection associated to parasitism by using populations as a different level of grouping, or are they looking for other signs of selection, possibly related to the ecology in different areas.
Table 4 may be placed in the Supplementary materials.
Author Response
Reviewer's comments in italics, with our response below.
In terms of the general structure of the manuscript, my main concern regards the possibility to provide a biological case example in support for the theoretical hypothesis that, in the competition between pararite and control agent, the one with the higher genetic variability will prevail. This is obviously presented as the driving rationale of the study, as it occupies the initial part of the abstract ad introduction and the conclusions. Nevertheless, in my view, this study cannot provide explicit support for the theory because there is no control, or baseline values, for variability and no data at all for the parasitoid. I would say that the observed high variability for the pest, the hypothesized low variability of the parasitoid and the observation that the parasitoid is loosing effectiveness after some generations are in line with the theory, but do not provide a nice and clean study case in its support. I would suggest presenting the study more simply as a genetic study of an important pest, not as a biological proof of the theory. The theory can obviously be introduced and discussed, but I do not see it as the main focus.
The change in title suggested by reviewer #1 partially addresses this comment. We agree with the reviewer that we have not provided biological proof of a resistance mechanism, and we have been careful not to imply that we have. To make this clearer, we have added the following sentence to the discussion: “More work will be required to find the genetic mechanism of resistance and describe its prevalence and spread in weevil populations, and to measure the amount of variation and population structure in M. hyperodae”. Without the material about variation in the host vs. the control agent, it’s not clear to us how we could provide a compelling rationale for our work.
Do authors have clues about the nature of the repeated sequences?
We ran RepeatClassifer on our (subset) draft assembly against Dfam 3.1. The majority of repeats were ‘Unclassified’. 9.2% of the genome was classified as Retroelements and 7.5% was classified as DNA transposons. We’ve added the results as a supplementary table.
Figures are complex and require extensive captions. Nevertheless captions now include quite some imformations that may be better placed in the methods or results sections.
We’ve arranged the manuscript so that the results presented in figures can be understood by reading the caption. We think moving this material to the methods would make the figures hard for readers to interpret.
The rationale behind testing for selection in parasitized/non parasitized pests is clear. Please clarify better the rationale for testing for selection between Northern and Southern populations once the former has proved ineffective. Are authors still looking for selection associated to parasitism by using populations as a different level of grouping, or are they looking for other signs of selection, possibly related to the ecology in different areas.
Detection of SNPs with skewed allele frequencies (by BayeScan) suggests selection, which could be related to ecology. We were also interested in the routes of entry into New Zealand and gene flow between populations. We have added the following sentence: “Regional genetic differences could be related to selection acting on different loci North and South of the Main Divide and/or genetic drift acting on isolated populations.”.
Table 4 may be placed in the Supplementary materials.
We’ve moved Table 4 to supplementary results.